

# Measuring the dispersion of rainfall using Bayesian confidence intervals for coefficient of variation of delta-lognormal distribution: a study from Thailand

Noppadon Yosboonruang, Sa-aat Niwitpong and Suparat Niwitpong

Department of Applied Statistics, Faculty of Applied Science, King Mongkut's University of
Technology North Bangkok, Bangkok, Thailand

## ABSTRACT

Since rainfall data series often contain zero values and thus follow a delta-lognormal distribution, the coefficient of variation is often used to illustrate the dispersion of rainfall in a number of areas and so is an important tool in statistical inference for a rainfall data series. Therefore, the aim in this paper is to establish new confidence intervals for a single coefficient of variation for delta-lognormal distributions using Bayesian methods based on the independent Jeffreys', the Jeffreys' Rule, and the uniform priors compared with the fiducial generalized confidence interval. The Bayesian methods are constructed with either equitailed confidence intervals or the highest posterior density interval. The performance of the proposed confidence intervals was evaluated using coverage probabilities and expected lengths via Monte Carlo simulations. The results indicate that the Bayesian equitailed confidence interval based on the independent Jeffreys' prior outperformed the other methods. Rainfall data recorded in national parks in July 2015 and in precipitation stations in August 2018 in Nan province, Thailand are used to illustrate the efficacy of the proposed methods using a real-life dataset.

## INTRODUCTION

Presently, the effects of global climate change caused by many factors, both natural and man-made (such as fuel burning, burning forests, deforestation, and oil drilling), are continuous. Such factors directly enhance natural changes such as the greenhouse effect and cause changes in precipitation, sea level, and the polar vortex. Thailand is a country that has been affected, as has been seen in the past few years. Especially in the north of Thailand, a lot of deforestation has caused flooding because there are insufficient trees to absorb water due to heavy rain. Subsequently, many organizations, both governmental and from the private sector, are interested in finding ways to mitigate the damage from such events, and thus a study on measuring the dispersion of rainfall in areas with the potential risk of flooding has become necessary. In statistics, the measurement of the

Corresponding author
Sa-aat Niwitpong,
sa-aat.n@sci.kmutnb.ac.th

coefficient of variation of rainfall data can illustrate the dispersion of rainfall and also predict the precipitation in each area as well. However, rainfall data are often zero-inflated, especially during the winter to summer months (October to May in Thailand; *Thai Meteorological Department, 2015*). Thus, the rainfall dataset follows a combination of two distributions: lognormal and binomial. Therefore, rainfall data follows a delta-lognormal distribution, as has been reported by many researchers (*Fukuchi, 1988*; *Shimizu, 1993*; *Yue, 2000*; *Kong et al., 2012*).

In many scientific studies, the data consist of positive right-skewed observations with an excess of exact zeros, and *Aitchison (1955)* determined that the distribution of this data is delta-lognormal, which has subsequently been used in other studies: for instance, the diagnostic charge data in *Callahan, Kesterson & Tierney (1997)* study (*Zhou & Tu, 2000*; *Li, Zhou & Tian, 2013*), fisheries data from a trawler survey carried out by the National Institute of Water and Atmospheric Research in New Zealand (*Fletcher, 2008*; *Wu & Hsieh, 2014*), household expenditure explored by the Ministry of Food in 1950 (*Aitchison, 1955*), and the concentration of airborne chlorine measured at an industrial site in the US (*Owen & DeRouen, 1980*; *Tian & Wu, 2006*).

To solve the problems when dealing with delta-lognormally distributed data, statistical inference is used which constructs confidence intervals for the parameters of interest, and in the past two decades, many researchers have investigated this. *Kvanli, Shen & Deng (1998)* constructed a confidence interval based on the likelihood ratio test approach for the population mean when there are many zeros in the data. *Zhou & Tu (2000)* proposed three different interval estimation procedures comprising a percentile-*t* bootstrap interval based on sufficient statistics and two likelihood-based confidence intervals for the mean of diagnostic test charge data containing zeros. *Chen & Zhou (2006)* introduced confidence intervals based on a true generalized pivotal (GP) method, an approximate GP method, a signed log-likelihood ratio (SLLR), and a modified SLLR for the ratio or difference between two means of lognormal populations with zeros. *Tian & Wu (2006)* constructed confidence intervals for the mean of lognormal data with excess zeros using an adjusted SLLR via an SLLR approach and a bootstrap approach. *Fletcher (2008)* used three methods, namely Aitchison's estimator, a modification of Cox's method for lognormal distributions, and a profile-likelihood interval to construct confidence intervals for the mean of a delta-lognormal distribution. *Buntao & Niwitpong (2013)* presented two confidence intervals: the concept of the GP approach (GPA) and the method of variance estimate recovery (MOVER) for the ratio of coefficients of variation of delta-lognormal distributions. *Li, Zhou & Tian (2013)* proposed two methods for the mean based on an approximate GP quantity and fiducial quantity of lognormal data with excess zeros. *Wu & Hsieh (2014)* established confidence intervals with Aitchison's method, a modified Land's method, the profile-likelihood interval, and the generalized confidence interval (GCI) for the mean of a delta-lognormal distribution. *Maneerat, Niwitpong & Niwitpong (2018)* introduced GCI, MOVER based on the variance stabilizing transformation (VST), the Wilson score interval, and Jeffreys' method to construct confidence intervals for the mean of a delta-lognormal distribution.

The coefficient of variation is another interesting parameter defined as the ratio of the standard deviation to the mean. It is useful to describe the dispersion of data and can be used to compare the degree of variation between two or more datasets with different measurement units. The coefficient of variation is used in several fields, such as medical science, meteorology, agriculture and economics (*Kim, Lee & Choi, 2005*; *Gulhar et al., 2012*; *Tian, 2005*). Recently, several researchers have considered various approaches to construct confidence intervals for coefficients of variation. For example, *Wong & Wu (2002)*, *Tian (2005)*, *Mahmoudvand & Hassani (2009)*, *Donner & Zou (2012)*, and *Wongkhao, Niwitpong & Niwitpong (2015)* established confidence intervals for the coefficient of variation for a normal distribution. After that, *Van Zyl & Van Der Merwe (2017)* proposed a Bayesian control chart for the common coefficient of variation for a normal distribution. In studies on two-parameter exponential distributions, *Sangnawakij & Niwitpong (2017)* used three methods, namely MOVER, GCI, and the asymptotic confidence interval, to establish confidence intervals for a single coefficient of variation and the difference between coefficients of variation, and *Thangjai & Niwitpong (2017)* presented confidence intervals based on an adjusted MOVER, GCI, and a large sample method for weighted coefficients of variation.

There have been studies on other skewed distributions, such as the one by *Fletcher (2008)* who presented three methods: Aitchison's estimator (the classical method), a modification of Cox's method for the lognormal, and a profile-likelihood interval, to construct confidence intervals for the mean of a delta-lognormal distribution. Fletcher suggested that Cox's method and profile-likelihood interval, which are the modified methods, are well performed to construct the confidence intervals for the mean of a delta-lognormal distribution. While Aitchison's estimator tend to have too low an upper limit. Therefore, Fletcher not recommend the Aitchison's estimator. *Buntao & Niwitpong (2012)* revealed the GPA and a closed-form method of variance estimation for coefficients of variation for both lognormal and delta-lognormal distributions. *Harvey & Van Der Merwe (2012)* constructed confidence intervals for means and variances of lognormal and bivariate lognormal distributions using a Bayesian method. *Niwitpong (2013)* presented a new confidence interval for the coefficient of variation of a lognormal distribution with restricted parameters. *D'Cunha & Rao (2014)* offered a Bayes confidence interval for the mean of a lognormal distribution and compared it with the maximum likelihood estimator method. *Sangnawakij, Niwitpong & Niwitpong (2015)* proposed MOVER with Score and Wald interval methods to construct confidence intervals for the ratio of coefficients of variation of gamma distributions. *Rao & D'Cunha (2016)* presented Bayesian confidence intervals for the median of a lognormal distribution and compared it with the confidence interval obtained from a Monte Carlo simulation. Recently, *Yosboonruang, Niwitpong & Niwitpong (2018)* constructed confidence intervals for the coefficient of variation of a delta-lognormal distribution based on a modified Fletcher method using the concept of *Fletcher (2008)*, and the GCI. The modified Fletcher, based on its variance, is the basic method to construct the confidence interval. Although this method failed in term of the coverage probability and the expected length, it is used to compare with the GCI. Moreover, they proposed methods including the fiducial generalized

confidence interval (FGCI) and MOVER based on the VST, the Wilson score, and Jeffreys' method to establish the confidence intervals for the coefficient of variation of three parameters of a delta-lognormal distribution, of which FGCI is recommended for constructing confidence intervals (*Yosboonruang, Niwitpong & Niwitpong, 2019*). In addition, they extended this study to construct confidence intervals for the coefficient of variation.

The goal of this study is to propose new confidence intervals using Bayesian methods and comparing them with FGCI proposed by *Yosboonruang, Niwitpong & Niwitpong (2019)* for a single coefficient of variation of a delta-lognormal distribution. The methods and theories to establish the confidence intervals are described in section "Methods". Next, a simulation study and results are presented in section "Results", and then the proposed methods are applied to the real-world datasets, as detailed in "An empirical study". The last two sections contain discussion and conclusions on the study.

## METHODS

Let $V = (V_1, V_2, \ldots, V_n)$ be a positive random variable from a lognormal distribution with parameters $\mu$ and $\sigma^2$, denoted as $\mathrm{LN}(\mu, \sigma^2)$. The probability density function of $V_i$ is given by

$$f(v_i; \mu, \sigma^2) = \begin{cases} \dfrac{1}{v_i \sigma \sqrt{2\pi}} \exp\left\{ -\dfrac{1}{2\sigma^2} [\ln(v_i) - \mu]^2 \right\} & ; \quad v_i > 0 \\ 0 & ; \quad \text{otherwise.} \end{cases} \tag{1}$$

Suppose that the population of interest contains both zero and non-zero observed values, denoted by $n_{(0)}$ and $n_{(1)}$, respectively, where $n = n_{(0)} + n_{(1)}$. The zero observations follow a binomial distribution, $n_{(0)} \sim \mathrm{Bin}(n, \delta')$, where $\delta' = 1 - \delta$ is the probability of zero observations, and the non-zero observations follow a lognormal distribution, thus resulting in a delta-lognormal distribution. Let $X = (X_1, X_2, \ldots, X_n)$ be a random sample from a delta-lognormal distribution, denoted by $\Delta(\delta', \mu, \sigma^2)$. The distribution function of a delta-lognormal population presented by *Tian & Wu (2006)* can be derived as

$$G(x_i; \delta', \mu, \sigma^2) = \begin{cases} \delta' & ; \quad x = 0 \\ \delta' + \delta F(x_i; \mu, \sigma^2) & ; \quad x > 0, \end{cases} \tag{2}$$

where $F(x_i; \mu, \sigma^2)$ is the lognormal cumulative distribution function. Let $Y_i = \ln(X_i) \sim N(\mu, \sigma^2)$ for $X_i > 0$. *Aitchison (1955)* described the respective population mean and variance of $X$ as

$$E(X) = \mu_X = \delta \exp\left( \mu + \frac{\sigma^2}{2} \right) \tag{3}$$

and

$$\mathrm{Var}(X) = \sigma_X^2 = \delta \exp(2\mu + \sigma^2)[\exp(\sigma^2) - \delta]. \tag{4}$$

The minimum variance unbiased estimator of $\mu_X$ was expressed by *Aitchison (1955)*; the estimator of $\mu_X$ is given by

$$\hat{\mu}_X = \hat{\delta} \exp(\hat{\mu}) \psi_{n_{(1)}} \left( \frac{\hat{\sigma}^2}{2} \right), \tag{5}$$

where $\hat{\delta} = \frac{n_{(1)}}{n}$, $\hat{\mu} = \frac{1}{n_{(1)}} \sum_{i=1}^{n_{(1)}} \ln(x_i)$, and $\hat{\sigma}^2 = \frac{1}{n_{(1)}-1} \sum_{i=1}^{n_{(1)}} [\ln(x_i) - \hat{\mu}]^2$, then the coefficient of variation of $X$ can be expressed as

$$\text{CV}(X) = \eta = \sqrt{\frac{\exp(\sigma^2)}{\delta} - 1}. \tag{6}$$

The methods to construct the confidence intervals for $\eta$ are proposed in the following section.

## The Bayesian confidence interval for a single coefficient of variation

If a delta-lognormal distribution has three unknown parameters $(\delta', \mu, \sigma^2)$, then the joint likelihood function is given by

$$L(\delta', \mu, \sigma^2 | \mathbf{x}) \propto (\delta')^{n_{(0)}} \delta^{n_{(1)}} \prod_{i=1}^{n_{(1)}} \frac{1}{\sqrt{\sigma^2}} \exp\left\{ -\frac{1}{2\sigma^2} [\ln(x_i) - \mu]^2 \right\}. \tag{7}$$

Therefore, the Fisher information matrix of the unknown parameters $(\delta', \mu, \sigma^2)$ per unit observation is written as

$$I(\delta', \mu, \sigma^2) = \begin{bmatrix} \frac{n}{\delta'\delta} & 0 & 0 \\ 0 & \frac{n\delta}{\sigma^2} & 0 \\ 0 & 0 & \frac{n\delta}{2(\sigma^2)^2} \end{bmatrix}. \tag{8}$$

In the following section, the Bayesian confidence interval is constructed upon three priors: the independent Jeffreys, Jeffreys' rule, and uniform.

## The Bayesian confidence interval using the independent Jeffreys' prior

Jeffreys' prior is defined as $p(\theta) \propto \sqrt{|I(\theta)|}$, where $I(\theta)$ is a Fisher information matrix. It is a non-informative prior distribution used in Bayesian parameter estimation and is very useful because it has the notable property of invariance under the reparameterization of $\theta$ (*Jeffreys, 1946*).

The independent Jeffreys' prior is a non-informative prior under the concept of establishing the product of Jeffreys' prior for each parameter while imposing staticity on the others (*Rubio & Liseo, 2014*).

For a binomial distribution, the parameter of interest is the probability $\delta'$, then the Jeffreys' invariant prior for a binomial parameter is given by

$$\begin{aligned} p(\delta') &\propto \sqrt{|I(\delta')|} \\ &\propto (\delta')^{-\frac{1}{2}} \delta^{-\frac{1}{2}}, \end{aligned} \tag{9}$$

which is $\text{Beta}(1/2, 1/2)$ (*Bolstad & Curran, 2017*). Subsequently, the posterior distribution of $\delta'$ is in the form

$$p(\delta' | n_{(0)}) \propto (\delta')^{n_{(0)} - \frac{1}{2}} \delta^{n_{(1)} - \frac{1}{2}}, \tag{10}$$

which is a beta distribution $\text{Beta}(n_{(0)} + 1/2, n_{(1)} + 1/2)$. Similarly, the independent Jeffreys' prior for a lognormal distribution is $p(\sigma^2) \propto \sigma^{-2}$. Therefore, the prior distribution for a delta-lognormal distribution can be expressed as

$$p(\delta', \sigma^2) \propto \sigma^{-2}(\delta')^{-\frac{1}{2}}\delta^{-\frac{1}{2}}. \tag{11}$$

The joint posterior density function is clearly defined as

$$p(\delta', \sigma^2|x) = \frac{1}{\text{Beta}(n_{(0)} + \frac{1}{2}, \ n_{(1)} + \frac{1}{2})}(\delta')^{n_{(0)} - \frac{1}{2}}\delta^{n_{(1)} - \frac{1}{2}} \times \frac{1}{\sqrt{2\pi}\frac{\sigma}{\sqrt{n_{(1)}}}}\exp\left[-\frac{1}{2\frac{\sigma^2}{n_{(1)}}}(\mu - \hat{\mu})^2\right]$$

$$\times \frac{\left[\frac{(n_{(1)} - 1)\hat{\sigma}^2}{2}\right]^{\frac{n_{(1)} - 1}{2}}}{\Gamma\left(\frac{n_{(1)} - 1}{2}\right)}(\sigma^2)^{-1 - \frac{(n_{(1)} - 1)}{2}}\exp\left[-\frac{(n_{(1)} - 1)\hat{\sigma}^2}{2\sigma^2}\right], \tag{12}$$

where $\hat{\mu} = \frac{1}{n_{(1)}}\sum_{i=1}^{n_{(1)}}\ln(x_i)$ and $\hat{\sigma}^2 = \frac{1}{n_{(1)} - 1}\sum_{i=1}^{n_{(1)}}[\ln(x_i) - \hat{\mu}]^2$. Since $\delta'$ and $\sigma^2$ are independent, then the posterior distributions of $\delta'$ and $\sigma^2$ are a beta and an inverse gamma distribution, respectively, as follows:

$$\delta'|x \sim \text{Beta}\left(n_{(0)} + \frac{1}{2}, n_{(1)} + \frac{1}{2}\right) \tag{13}$$

and

$$p(\sigma^2|x) = \frac{\left[\frac{(n_{(1)} - 1)\hat{\sigma}^2}{2}\right]^{\frac{n_{(1)} - 1}{2}}}{\Gamma\left(\frac{n_{(1)} - 1}{2}\right)}(\sigma^2)^{-1 - \frac{(n_{(1)} - 1)}{2}}\exp\left[-\frac{(n_{(1)} - 1)\hat{\sigma}^2}{2\sigma^2}\right]. \tag{14}$$

To construct the Bayesian confidence interval, $\delta$ and $\sigma^2$ in Eq. (6) are substituted by $\delta' \mid x$ and $\sigma^2 \mid x$ defined in Eqs. (13) and (14), respectively. Therefore, the $100(1 - \alpha)\%$ two-sided confidence interval for the coefficient of variation based on the independent Jeffreys' prior Bayesian is obtained by

$$CI_\eta^{\text{B.indj}} = \left[L_\eta^{\text{B.indj}}, U_\eta^{\text{B.indj}}\right], \tag{15}$$

where $L_\eta^{\text{B.indj}}$ and $U_\eta^{\text{B.indj}}$ are the lower and upper bounds of the $100(1 - \alpha)\%$ equitailed confidence interval and the highest posterior density (HPD) interval of $\eta$, respectively.

The HPD interval is an interval in the domain of a posterior probability distribution which gives the narrowest length of the interval (*Hyndman, 1995*; *Yau & Campbell, 2019*). It represents the most credible points which cover most of the distribution. In addition, each point inside the interval has a higher probability density than those outside it.

## The Bayesian confidence interval using the Jeffreys' Rule prior

As mentioned previously, the Jeffreys' Rule prior is obtained from the square root of the determinant of the Fisher information matrix. This prior is appropriate for a single parameter. The Jeffreys' Rule prior has the rule that the prior is invariant (the valuable property) (*Lee, 2012*), which is imposed as $p(\sigma^2) \propto \sigma^{-3}$. From *Harvey & Van Der Merwe (2012)*,

the Jeffreys' Rule prior for $\delta'$ in a binomial distribution is $p(\delta') \propto (\delta')^{-\frac{1}{2}} \delta^{\frac{1}{2}}$. It is easy to find the Jeffreys' Rule prior for the delta-lognormal distribution, which is defined as

$$p(\delta', \sigma^2) \propto \sigma^{-3}(\delta')^{-\frac{1}{2}} \delta^{\frac{1}{2}}. \tag{16}$$

Subsequently, the joint posterior density is given by

$$p(\delta', \sigma^2 | x) = \frac{1}{\text{Beta}\left(n_{(0)} + \frac{1}{2}, n_{(1)} + \frac{3}{2}\right)} (\delta')^{n_{(0)} - \frac{1}{2}} \delta^{n_{(1)} + \frac{1}{2}} \times \frac{1}{\sqrt{2\pi} \frac{\sigma}{\sqrt{n_{(1)}}}} \exp\left[-\frac{1}{2\frac{\sigma^2}{n_{(1)}}}(\mu - \hat{\mu})^2\right]$$

$$\times \frac{\left[\frac{n_{(1)}\hat{\sigma}^2}{2}\right]^{\frac{n_{(1)}}{2}}}{\Gamma\left(\frac{n_{(1)}}{2}\right)} (\sigma^2)^{-1 - \frac{n_{(1)}}{2}} \exp\left[-\frac{n_{(1)}\hat{\sigma}^2}{2\sigma^2}\right], \tag{17}$$

where $\hat{\mu} = \frac{1}{n_{(1)}} \sum_{i=1}^{n_{(1)}} \ln(x_i)$ and $\hat{\sigma}^2 = \frac{1}{n_{(1)} - 1} \sum_{i=1}^{n_{(1)}} [\ln(x_i) - \hat{\mu}]^2$. In addition, the posterior density of $\delta'$ becomes

$$\delta' | x \sim \text{Beta}\left(n_{(0)} + \frac{1}{2}, n_{(1)} + \frac{3}{2}\right) \tag{18}$$

and the posterior distribution of $\sigma^2$ can be expressed as

$$p(\sigma^2 | x) = \frac{\left[\frac{n_{(1)}\hat{\sigma}^2}{2}\right]^{\frac{n_{(1)}}{2}}}{\Gamma\left(\frac{n_{(1)}}{2}\right)} (\sigma^2)^{-1 - \frac{n_{(1)}}{2}} \exp\left[-\frac{n_{(1)}\hat{\sigma}^2}{2\sigma^2}\right]. \tag{19}$$

Next, the confidence limit of $\eta$ is constructed using $\delta' \mid x$ and $\sigma^2 \mid x$ given by Eqs. (18) and (19), respectively. Therefore, the $100(1 - \alpha)\%$ equitailed confidence interval and HPD interval for the coefficient of variation based on the Jeffreys' Rule prior Bayesian are obtained by

$$CI_\eta^{\text{B.jrule}} = \left[L_\eta^{\text{B.jrule}}, U_\eta^{\text{B.jrule}}\right], \tag{20}$$

where $L_\eta^{\text{B.jrule}}$ and $U_\eta^{\text{B.jrule}}$ are the lower and upper bounds of the confidence limit, respectively.

### The Bayesian confidence interval using the uniform prior

The prior probability of the uniform prior is a constant function (*Stone, 2013*). This means that the uniform prior gives equally likely a priori to all possible values (*O'Reilly & Mars, 2015*). The uniform prior for the binomial proportion is $p(\delta') \propto 1$ (*Bolstad & Curran, 2017*), that for $\sigma^2$ is $p(\sigma^2) \propto 1$ (*Kalkur & Rao, 2017*), and that of a delta-lognormal distribution is $p(\delta', \sigma^2) \propto 1$. The joint posterior density function can be expressed as

$$p(\delta', \sigma^2 | x) = \frac{1}{\text{Beta}\left(n_{(0)} + 1, n_{(1)} + 1\right)} (\delta')^{n_{(0)}} \delta^{n_{(1)}} \frac{1}{\sqrt{2\pi} \frac{\sigma}{\sqrt{n_{(1)}}}} \exp\left[-\frac{1}{2\frac{\sigma^2}{n_{(1)}}}(\mu - \hat{\mu})^2\right]$$

$$\times \frac{\left[\frac{(n_{(1)} - 2)\hat{\sigma}^2}{2}\right]^{\frac{n_{(1)} - 2}{2}}}{\Gamma\left(\frac{n_{(1)} - 2}{2}\right)} (\sigma^2)^{-1 - \frac{(n_{(1)} - 2)}{2}} \exp\left[-\frac{(n_{(1)} - 2)\hat{\sigma}^2}{2\sigma^2}\right], \tag{21}$$

| Algorithm 1 |
| --- |

*Step 1*: Generate $\mathbf{x}_i$, $i = 1, 2, ..., n$ from a delta-lognormal distribution.

*Step 2*: Compute $\hat{\delta}$ and $\hat{\sigma}^2$.

*Step 3*: Generate $\delta' \mid \mathbf{x}$, which is the beta distribution from Eqs. (13), (18), and (22).

*Step 4*: Generate $\sigma^2 \mid \mathbf{x}$, which is the inverse gamma distribution from Eqs. (14), (19), and (23).

*Step 5*: Compute $\eta$ by substituting $\delta' \mid \mathbf{x}$ and $\sigma^2 \mid \mathbf{x}$ in Eq. (6).

*Step 6*: Repeat Steps 3–5 5,000 times and obtain an array of $\eta$.

*Step 7*: Compute the 95% equitailed confidence interval and HPD interval for $\eta$ from Eqs. (15), (20), and (24). If $L \leq \eta \leq U$, then set $cp = 1$; else, set $cp = 0$.

*Step 8*: Repeat Steps 1–7 15,000 times to compute the coverage probability and the expected length.

where $\hat{\mu} = \frac{1}{n_{(1)}} \sum_{i=1}^{n_{(1)}} \ln(x_i)$ and $\hat{\sigma}^2 = \frac{1}{n_{(1)}-1} \sum_{i=1}^{n_{(1)}} [\ln(x_i) - \hat{\mu}]^2$. By Eq. (21), the respective posterior distributions of $\delta'$ and $\sigma^2$ are formed as

$$\delta'|x \sim \text{Beta}\left(n_{(0)} + 1, n_{(1)} + 1\right) \tag{22}$$

and

$$p\left(\sigma^2|x\right) = \frac{\left[\frac{\left(n_{(1)}-2\right)\hat{\sigma}^2}{2}\right]^{\frac{n_{(1)}-2}{2}}}{\Gamma\left(\frac{n_{(1)}-2}{2}\right)} \left(\sigma^2\right)^{-1-\frac{\left(n_{(1)}-2\right)}{2}} \exp\left[-\frac{\left(n_{(1)}-2\right)\hat{\sigma}^2}{2\sigma^2}\right], \tag{23}$$

which are beta and inverse gamma distributions, respectively. From Eqs. (22) and (23), the confidence limit for $\eta$ can be established, and consequently, the $100(1 - \alpha)\%$ equitailed confidence interval and HPD interval for the coefficient of variation based on the uniform prior Bayesian are as follows:

$$CI_{\eta}^{\text{B.uni}} = \left[L_{\eta}^{\text{B.uni}}, U_{\eta}^{\text{B.uni}}\right], \tag{24}$$

where $L_{\eta}^{\text{B.uni}}$ and $U_{\eta}^{\text{B.uni}}$ are the lower and upper bounds of the confidence limit, respectively.

## The FGCI for a single coefficient of variation

The fiducial approach was first introduced by *Fisher (1930)*, after which it has been used to construct confidence limits by many researchers, such as *Hannig, Abdel-Karim & Iyer (2006)*, *Hannig, Iyer & Patterson (2006)*, *Hannig (2009)*, *Hannig & Lee (2009)*, *Li, Zhou & Tian (2013)*, and *Yosboonruang, Niwitpong & Niwitpong (2019)*. The concept of FGCI uses the respective generalized fiducial quantities for $\delta$ and $\sigma^2$ (*Li, Zhou & Tian, 2013*):

$$R_{\delta} \sim \frac{1}{2}\text{Beta}\left(n_{(1)}, n_{(0)} + 1\right) + \frac{1}{2}\text{Beta}\left(n_{(1)} + 1, n_{(0)}\right) \tag{25}$$

and

$$R_{\sigma^2} = \frac{\left(n_{(1)} - 1\right)\hat{\sigma}^2}{U}, \tag{26}$$

**Algorithm 2**

*Step 1*: Generate $x_i$, $i = 1, 2, ..., n$ from a delta-lognormal distribution.

*Step 2*: Compute $\hat{\delta}$ and $\hat{\sigma}^2$.

*Step 3*: Generate $\text{Beta}(n_{(1)}, n_{(0)} + 1)$ and $\text{Beta}(n_{(1)} + 1, n_{(0)})$.

*Step 4*: Compute $R_\delta$, $R_{\sigma 2}$, and $R_\eta$ from Eqs. (25), (26), and (27), respectively.

*Step 5*: Repeat Steps 3–4 5,000 times and obtain an array of $R_\eta$.

*Step 6*: Compute the 95% confidence intervals for $\eta$ from Eq. (28). If $L \leq \eta \leq U$, then set $cp = 1$; else, set $cp = 0$.

*Step 7*: Repeat Steps 1–6 15,000 times to compute the coverage probability and the expected length.

where $U \sim \chi^2_{n_{(1)}-1}$. Subsequently, the generalized fiducial quantity for $\eta$ is

$$R_\eta = \sqrt{\frac{\exp(R_{\sigma 2})}{R_\delta} - 1}. \tag{27}$$

Therefore, the $100(1 - \alpha)\%$ generalized fiducial quantity interval for the coefficient of variation is defined by

$$CI_\eta^{\text{fgci}} = \left[ R_\eta(\alpha/2), R_\eta(1 - \alpha/2) \right], \tag{28}$$

where $R_\eta(\alpha/2)$ and $R_\eta(1 - \alpha/2)$ are the $100(\alpha/2)$-th and $100(1 - \alpha/2)$-th percentiles of the distribution of $R_\eta$, respectively.

## RESULTS

To evaluate the performance of the proposed methods, their coverage probabilities and expected lengths were estimated via Monte Carlo simulation using the R statistical programming language (*Venables & Smith, 2009*). Normally, the best confidence intervals are chosen from the coverage probability that is greater than or closest to the nominal confidence level and has the shortest expected lengths. In the simulation study, sample size $n$ was set as 25, 50, 100, 200; $\mu$ as 0; $\delta$ as 0.2, 0.5, 0.8, 0.9; and $\sigma^2$ as 0.1, 0.5, 1.0, 2.0. We eliminated the case of $n = 25$, $\delta = 0.2$ and $\sigma^2 = 0.1, 0.5, 1.0, 2.0$ because the expected non-zero observations were less than 10 (see *Fletcher, 2008*; *Wu & Hsieh, 2014*). For all of the simulations, the number of replications was set as 15,000, and 5,000 repetitions were used for the Bayesian and FGCI methods; the nominal confidence level was 0.95.

The results in Table 1 show that the Bayesian method using the independent Jeffreys' prior for the equitailed confidence interval outperformed the others because the coverage probabilities were consistently greater than or close to the target in all cases. In addition, for the equitailed confidence intervals, the coverage probabilities of the Bayesian using the Jeffreys' Rule prior were less than the nominal confidence level of 0.95 for some of the cases: $n = 25$, $\delta = 0.5$, $\sigma^2 = 0.1, 2.0$; $n = 50, 100$, $\delta = 0.2$, $\sigma^2 = 0.1, 2.0$; and $n = 200$, $\delta = 0.2$, $\sigma^2 = 0.1$. For the Bayesian method using the uniform prior, the coverage probabilities were close to 1 in a few cases when the sample sizes were less than 100 and had small variances together with high proportion of non-zero values. For the method with HPD intervals, the coverage probabilities of the independent Jeffreys' prior

**Table 1 The coverage probabilities of 95% two-sided confidence intervals for a single coefficient of variation with the delta-lognormal distribution.**

| n | δ | σ² | Coverage probabilities | | | | | | |
|---|---|---|---|---|---|---|---|---|---|
| | | | Equitailed confidence intervals | | | HPD intervals | | | FGCI |
| | | | Independent Jeffreys | Jeffreys' Rule | Uniform | Independent Jeffreys | Jeffreys' Rule | Uniform | |
| 25 | 0.5 | 0.1 | 0.9600 | 0.9397 | 0.9647 | 0.9413 | 0.9181 | 0.9475 | 0.8686 |
| | | 0.5 | 0.9718 | 0.9595 | 0.9763 | 0.9526 | 0.9308 | 0.9591 | 0.9415 |
| | | 1.0 | 0.9593 | 0.9481 | 0.9668 | 0.9381 | 0.9187 | 0.9461 | 0.9518 |
| | | 2.0 | 0.9521 | 0.9438 | 0.9608 | 0.9421 | 0.9309 | 0.9505 | 0.9523 |
| | 0.8 | 0.1 | 0.9721 | 0.9657 | 0.9848 | 0.9539 | 0.9446 | 0.9746 | 0.9192 |
| | | 0.5 | 0.9677 | 0.9618 | 0.9733 | 0.9579 | 0.9498 | 0.9686 | 0.9541 |
| | | 1.0 | 0.9541 | 0.9487 | 0.9601 | 0.9499 | 0.9413 | 0.9589 | 0.9512 |
| | | 2.0 | 0.9533 | 0.9482 | 0.9583 | 0.9482 | 0.9407 | 0.9551 | 0.9506 |
| | 0.9 | 0.1 | 0.9669 | 0.9610 | 0.9960 | 0.9439 | 0.9391 | 0.9865 | 0.9393 |
| | | 0.5 | 0.9607 | 0.9553 | 0.9683 | 0.9565 | 0.9500 | 0.9697 | 0.9559 |
| | | 1.0 | 0.9511 | 0.9463 | 0.9573 | 0.9521 | 0.9464 | 0.9618 | 0.9539 |
| | | 2.0 | 0.9524 | 0.9467 | 0.9563 | 0.9532 | 0.9475 | 0.9596 | 0.9481 |
| 50 | 0.2 | 0.1 | 0.9622 | 0.9349 | 0.9569 | 0.9384 | 0.9029 | 0.9289 | 0.8687 |
| | | 0.5 | 0.9741 | 0.9547 | 0.9729 | 0.9539 | 0.9269 | 0.9517 | 0.9403 |
| | | 1.0 | 0.9641 | 0.9477 | 0.9684 | 0.9449 | 0.9175 | 0.9477 | 0.9499 |
| | | 2.0 | 0.9553 | 0.9447 | 0.9641 | 0.9402 | 0.9203 | 0.9485 | 0.9491 |
| | 0.5 | 0.1 | 0.9605 | 0.9476 | 0.9619 | 0.9471 | 0.9301 | 0.9504 | 0.8694 |
| | | 0.5 | 0.9669 | 0.9579 | 0.9689 | 0.9563 | 0.9435 | 0.9586 | 0.9356 |
| | | 1.0 | 0.9585 | 0.9521 | 0.9622 | 0.9446 | 0.9329 | 0.9490 | 0.9499 |
| | | 2.0 | 0.9534 | 0.9485 | 0.9575 | 0.9426 | 0.9346 | 0.9462 | 0.9537 |
| | 0.8 | 0.1 | 0.9651 | 0.9600 | 0.9755 | 0.9508 | 0.9436 | 0.9659 | 0.9018 |
| | | 0.5 | 0.9623 | 0.9581 | 0.9671 | 0.9547 | 0.9486 | 0.9621 | 0.9495 |
| | | 1.0 | 0.9557 | 0.9523 | 0.9590 | 0.9467 | 0.9421 | 0.9527 | 0.9523 |
| | | 2.0 | 0.9551 | 0.9529 | 0.9574 | 0.9525 | 0.9488 | 0.9556 | 0.9487 |
| | 0.9 | 0.1 | 0.9660 | 0.9640 | 0.9830 | 0.9515 | 0.9463 | 0.9720 | 0.9274 |
| | | 0.5 | 0.9581 | 0.9555 | 0.9623 | 0.9543 | 0.9509 | 0.9621 | 0.9537 |
| | | 1.0 | 0.9519 | 0.9496 | 0.9554 | 0.9474 | 0.9443 | 0.9535 | 0.9535 |
| | | 2.0 | 0.9507 | 0.9484 | 0.9537 | 0.9483 | 0.9450 | 0.9518 | 0.9501 |
| 100 | 0.2 | 0.1 | 0.9571 | 0.9355 | 0.9513 | 0.9406 | 0.9155 | 0.9325 | 0.8582 |
| | | 0.5 | 0.9673 | 0.9532 | 0.9655 | 0.9539 | 0.9350 | 0.9509 | 0.9288 |
| | | 1.0 | 0.9612 | 0.9519 | 0.9623 | 0.9433 | 0.9263 | 0.9430 | 0.9461 |
| | | 2.0 | 0.9511 | 0.9435 | 0.9563 | 0.9401 | 0.9279 | 0.9434 | 0.9491 |
| | 0.5 | 0.1 | 0.9578 | 0.9462 | 0.9587 | 0.9473 | 0.9356 | 0.9487 | 0.8591 |
| | | 0.5 | 0.9605 | 0.9531 | 0.9621 | 0.9531 | 0.9432 | 0.9543 | 0.9315 |
| | | 1.0 | 0.9566 | 0.9532 | 0.9585 | 0.9433 | 0.9367 | 0.9455 | 0.9468 |
| | | 2.0 | 0.9546 | 0.9518 | 0.9559 | 0.9457 | 0.9412 | 0.9472 | 0.9516 |

| Table 1 (continued). | | | | | | | | |
|---|---|---|---|---|---|---|---|---|
| $n$ | $\delta$ | $\sigma^2$ | Coverage probabilities | | | | | |
| | | | Equitailed confidence intervals | | | HPD intervals | | | FGCI |
| | | | Independent Jeffreys | Jeffreys' Rule | Uniform | Independent Jeffreys | Jeffreys' Rule | Uniform | |
| | 0.8 | 0.1 | 0.9603 | 0.9566 | 0.9694 | 0.9464 | 0.9413 | 0.9578 | 0.8845 |
| | | 0.5 | 0.9605 | 0.9580 | 0.9641 | 0.9461 | 0.9425 | 0.9508 | 0.9442 |
| | | 1.0 | 0.9533 | 0.9505 | 0.9544 | 0.9485 | 0.9473 | 0.9517 | 0.9514 |
| | | 2.0 | 0.9509 | 0.9499 | 0.9524 | 0.9476 | 0.9458 | 0.9499 | 0.9509 |
| | 0.9 | 0.1 | 0.9657 | 0.9633 | 0.9768 | 0.9495 | 0.9461 | 0.9667 | 0.9065 |
| | | 0.5 | 0.9538 | 0.9533 | 0.9582 | 0.9544 | 0.9509 | 0.9601 | 0.9473 |
| | | 1.0 | 0.9534 | 0.9507 | 0.9539 | 0.9491 | 0.9475 | 0.9529 | 0.9491 |
| | | 2.0 | 0.9505 | 0.9493 | 0.9527 | 0.9498 | 0.9480 | 0.9509 | 0.9512 |
| 200 | 0.2 | 0.1 | 0.9565 | 0.9407 | 0.9513 | 0.9425 | 0.9263 | 0.9381 | 0.8541 |
| | | 0.5 | 0.9591 | 0.9473 | 0.9570 | 0.9485 | 0.9355 | 0.9461 | 0.9189 |
| | | 1.0 | 0.9577 | 0.9496 | 0.9577 | 0.9406 | 0.9281 | 0.9397 | 0.9423 |
| | | 2.0 | 0.9561 | 0.9515 | 0.9567 | 0.9399 | 0.9326 | 0.9417 | 0.9477 |
| | 0.5 | 0.1 | 0.9564 | 0.9490 | 0.9573 | 0.9463 | 0.9389 | 0.9483 | 0.8567 |
| | | 0.5 | 0.9565 | 0.9517 | 0.9575 | 0.9507 | 0.9440 | 0.9508 | 0.9249 |
| | | 1.0 | 0.9555 | 0.9531 | 0.9560 | 0.9433 | 0.9397 | 0.9439 | 0.9438 |
| | | 2.0 | 0.9541 | 0.9521 | 0.9552 | 0.9445 | 0.9408 | 0.9461 | 0.9469 |
| | 0.8 | 0.1 | 0.9575 | 0.9537 | 0.9625 | 0.9509 | 0.9454 | 0.9575 | 0.8771 |
| | | 0.5 | 0.9555 | 0.9531 | 0.9578 | 0.9506 | 0.9476 | 0.9536 | 0.9409 |
| | | 1.0 | 0.9517 | 0.9503 | 0.9523 | 0.9475 | 0.9465 | 0.9497 | 0.9503 |
| | | 2.0 | 0.9507 | 0.9500 | 0.9513 | 0.9464 | 0.9457 | 0.9473 | 0.9527 |
| | 0.9 | 0.1 | 0.9603 | 0.9587 | 0.9693 | 0.9495 | 0.9465 | 0.9601 | 0.8949 |
| | | 0.5 | 0.9541 | 0.9527 | 0.9565 | 0.9482 | 0.9463 | 0.9505 | 0.9445 |
| | | 1.0 | 0.9523 | 0.9518 | 0.9531 | 0.9462 | 0.9444 | 0.9468 | 0.9480 |
| | | 2.0 | 0.9513 | 0.9513 | 0.9517 | 0.9514 | 0.9511 | 0.9523 | 0.9500 |

did not cover the target in most cases, especially for large sample sizes. Similarly, a few cases with the Bayesian method using the uniform prior had coverage probabilities less than the nominal confidence level when the sample sizes were large. Moreover, the Bayesian method using the Jeffreys' Rule prior had coverage probabilities of less than 0.95 in almost all cases. Last, the coverage probabilities with FGCI did not cover the nominal confidence level when the variances were small for all sample sizes. In addition, when considering the expected lengths of all methods which is shown in Table 2, these were wide in cases of $\sigma^2 = 2.0$ and became narrower when the sample size increased, although they corresponded with the coverage probabilities in almost all cases. Furthermore, the values were similar for all of the methods. Moreover, the expected lengths of the interval when $n = 50$, $\delta = 0.2$, and $\sigma^2 = 2.0$ were very much larger than the other cases because the number of expected non-zero observations was small together with a large variance. This case might
**Table 2 The expected lengths of 95% two-sided confidence intervals for a single coefficient of variation with the delta-lognormal distribution.**

| $n$ | $\delta$ | $\sigma^2$ | Expected lengths | | | | | | FGCI |
|---|---|---|---|---|---|---|---|---|---|
| | | | Equitailed confidence intervals | | | HPD intervals | | | |
| | | | Independent Jeffreys | Jeffreys' Rule | Uniform | Independent Jeffreys | Jeffreys' Rule | Uniform | |
| 25 | 0.5 | 0.1 | 0.7297 | 0.6901 | 0.7259 | 0.7126 | 0.6750 | 0.7087 | 0.5254 |
| | | 0.5 | 1.5150 | 1.4019 | 1.6064 | 1.3664 | 1.2766 | 1.4273 | 1.3969 |
| | | 1.0 | 4.2513 | 3.8066 | 4.7342 | 3.2613 | 2.9912 | 3.5216 | 4.1004 |
| | | 2.0 | 33.7300 | 27.5715 | 42.9352 | 17.2204 | 14.9908 | 19.8622 | 32.8632 |
| | 0.8 | 0.1 | 0.4319 | 0.4176 | 0.4390 | 0.4222 | 0.4084 | 0.4296 | 0.3280 |
| | | 0.5 | 0.8803 | 0.8469 | 0.9149 | 0.8168 | 0.7885 | 0.8457 | 0.8324 |
| | | 1.0 | 2.1046 | 2.0054 | 2.2163 | 1.7993 | 1.7294 | 1.8798 | 2.0750 |
| | | 2.0 | 9.4926 | 8.8318 | 10.2578 | 6.9057 | 6.5518 | 7.3427 | 9.4872 |
| | 0.9 | 0.1 | 0.3346 | 0.3248 | 0.3518 | 0.3232 | 0.3139 | 0.3410 | 0.2710 |
| | | 0.5 | 0.7577 | 0.7342 | 0.7880 | 0.7082 | 0.6881 | 0.7346 | 0.7376 |
| | | 1.0 | 1.7631 | 1.6978 | 1.8460 | 1.5534 | 1.5049 | 1.6173 | 1.7591 |
| | | 2.0 | 7.4883 | 7.0855 | 7.9907 | 5.8011 | 5.5645 | 6.1147 | 7.4510 |
| 50 | 0.2 | 0.1 | 1.3339 | 1.2287 | 1.3061 | 1.2863 | 1.1889 | 1.2586 | 0.9404 |
| | | 0.5 | 3.0407 | 2.6749 | 3.3153 | 2.5845 | 2.3218 | 2.7216 | 2.7064 |
| | | 1.0 | 10.4439 | 8.5127 | 12.8712 | 6.8066 | 5.8704 | 7.6676 | 9.7812 |
| | | 2.0 | 218.3340 | 123.9037 | 409.4769 | 77.4546 | 55.6155 | 141.4522 | 517.7059 |
| | 0.5 | 0.1 | 0.5278 | 0.5128 | 0.5246 | 0.5204 | 0.5059 | 0.5174 | 0.3787 |
| | | 0.5 | 0.9170 | 0.8878 | 0.9311 | 0.8739 | 0.8476 | 0.8849 | 0.8188 |
| | | 1.0 | 2.0182 | 1.9446 | 2.0759 | 1.8142 | 1.7545 | 1.8557 | 1.9373 |
| | | 2.0 | 7.8105 | 7.4368 | 8.1546 | 6.3278 | 6.0759 | 6.5403 | 7.9104 |
| | 0.8 | 0.1 | 0.3063 | 0.3012 | 0.3082 | 0.3023 | 0.2972 | 0.3043 | 0.2298 |
| | | 0.5 | 0.5519 | 0.5429 | 0.5603 | 0.5349 | 0.5265 | 0.5427 | 0.5203 |
| | | 1.0 | 1.1725 | 1.1514 | 1.1951 | 1.0952 | 1.0773 | 1.1143 | 1.1529 |
| | | 2.0 | 3.9949 | 3.9093 | 4.0925 | 3.5027 | 3.4371 | 3.5760 | 3.9755 |
| | 0.9 | 0.1 | 0.2436 | 0.2400 | 0.2491 | 0.2390 | 0.2355 | 0.2447 | 0.1923 |
| | | 0.5 | 0.4867 | 0.4799 | 0.4948 | 0.4717 | 0.4655 | 0.4793 | 0.4729 |
| | | 1.0 | 1.0405 | 1.0248 | 1.0588 | 0.9804 | 0.9674 | 0.9971 | 1.7590 |
| | | 2.0 | 3.4585 | 3.3987 | 3.5352 | 3.0534 | 3.0069 | 3.1113 | 3.4754 |
| 100 | 0.2 | 0.1 | 0.9341 | 0.8971 | 0.9200 | 0.9190 | 0.8829 | 0.9050 | 0.6628 |
| | | 0.5 | 1.6376 | 1.5617 | 1.6550 | 1.5542 | 1.4859 | 1.5640 | 1.4196 |
| | | 1.0 | 3.7403 | 3.5325 | 3.8579 | 3.2758 | 3.1139 | 3.3492 | 3.5025 |
| | | 2.0 | 16.4708 | 15.2363 | 17.4752 | 12.4579 | 11.7196 | 12.9891 | 15.9392 |
| | 0.5 | 0.1 | 0.3774 | 0.3719 | 0.3760 | 0.3738 | 0.3685 | 0.3725 | 0.2712 |
| | | 0.5 | 0.6066 | 0.5975 | 0.6096 | 0.5957 | 0.5871 | 0.5986 | 0.5354 |
| | | 1.0 | 1.2286 | 1.2092 | 1.2409 | 1.1698 | 1.1521 | 1.1801 | 1.1719 |
| | | 2.0 | 3.9868 | 3.9149 | 4.0450 | 3.6197 | 3.5622 | 3.6637 | 3.9565 |

| n | δ | σ² | Expected lengths | | | | | | |
|---|---|---|---|---|---|---|---|---|---|
| | | | Equitailed confidence intervals | | | HPD intervals | | | FGCI |
| | | | Independent Jeffreys | Jeffreys' Rule | Uniform | Independent Jeffreys | Jeffreys' Rule | Uniform | |
| | 0.8 | 0.1 | 0.2190 | 0.2172 | 0.2196 | 0.2171 | 0.2153 | 0.2177 | 0.1633 |
| | | 0.5 | 0.3732 | 0.3703 | 0.3758 | 0.3664 | 0.3637 | 0.3689 | 0.3487 |
| | | 1.0 | 0.7565 | 0.7502 | 0.7625 | 0.7330 | 0.7274 | 0.7388 | 0.7426 |
| | | 2.0 | 2.3285 | 2.3082 | 2.3515 | 2.1759 | 2.1582 | 2.1948 | 2.3232 |
| | 0.9 | 0.1 | 0.1743 | 0.1729 | 0.1761 | 0.1723 | 0.1710 | 0.1742 | 0.1358 |
| | | 0.5 | 0.3296 | 0.3275 | 0.3322 | 0.3236 | 0.3216 | 0.3262 | 0.3182 |
| | | 1.0 | 0.6764 | 0.6720 | 0.6817 | 0.6560 | 0.6521 | 0.6612 | 0.6717 |
| | | 2.0 | 2.0505 | 2.0360 | 2.0685 | 1.9451 | 1.9328 | 1.9618 | 2.0511 |
| 200 | 0.2 | 0.1 | 0.6714 | 0.6576 | 0.6658 | 0.6638 | 0.6502 | 0.6582 | 0.4777 |
| | | 0.5 | 1.0731 | 1.0499 | 1.0742 | 1.0478 | 1.0255 | 1.0481 | 0.9122 |
| | | 1.0 | 2.1624 | 2.1109 | 2.1802 | 2.0517 | 2.0062 | 2.0653 | 2.0429 |
| | | 2.0 | 7.4107 | 7.2073 | 7.5181 | 6.5125 | 6.3549 | 6.5886 | 7.2846 |
| | 0.5 | 0.1 | 0.2704 | 0.2684 | 0.2699 | 0.2685 | 0.2666 | 0.2680 | 0.1946 |
| | | 0.5 | 0.4194 | 0.4163 | 0.4202 | 0.4154 | 0.4123 | 0.4161 | 0.3676 |
| | | 1.0 | 0.8190 | 0.8128 | 0.8224 | 0.7971 | 0.7910 | 0.8001 | 0.7794 |
| | | 2.0 | 2.4764 | 2.4567 | 2.4900 | 2.3551 | 2.3375 | 2.3669 | 2.4492 |
| | 0.8 | 0.1 | 0.1566 | 0.1559 | 0.1568 | 0.1557 | 0.1550 | 0.1558 | 0.1163 |
| | | 0.5 | 0.2587 | 0.2577 | 0.2595 | 0.2562 | 0.2552 | 0.2569 | 0.2415 |
| | | 1.0 | 0.5145 | 0.5127 | 0.5166 | 0.5055 | 0.5036 | 0.5073 | 0.5045 |
| | | 2.0 | 1.5141 | 1.5085 | 1.5205 | 1.4683 | 1.4623 | 1.4744 | 1.5080 |
| | 0.9 | 0.1 | 0.1247 | 0.1243 | 0.1254 | 0.1238 | 0.1233 | 0.1244 | 0.0965 |
| | | 0.5 | 0.2283 | 0.2276 | 0.2292 | 0.2260 | 0.2254 | 0.2268 | 0.2200 |
| | | 1.0 | 0.4617 | 0.4602 | 0.4635 | 0.4521 | 0.4508 | 0.4539 | 0.4557 |
| | | 2.0 | 1.3467 | 1.3423 | 1.3523 | 1.3067 | 1.3028 | 1.3121 | 1.3467 |

have affected the parameter estimation, thus it is possible that the efficacy of the confidence intervals constructed from it was not very good.

## An empirical study

*Ananthakrishnan & Soman (1989)* studied a daily rainfall data series focusing on the normalized rainfall curve (NRC). They found that the NRC is uniquely determined by the coefficient of variation of the rainfall series. To verify the effectiveness of the proposed confidence intervals, we used two examples of rainfall datasets from Nan province, Thailand as follows.

### Example 1

The rainfall data was collected in July 2015 for national parks in Nan province, Thailand: Doi Phu Kha, Mae Charim, Nanthaburi, Tham Sa Koen, Sri Nan, Khun Sathan, and Doi Pha Klong recorded by the Protected Area Regional Office 13 Phrae, Thailand. For this

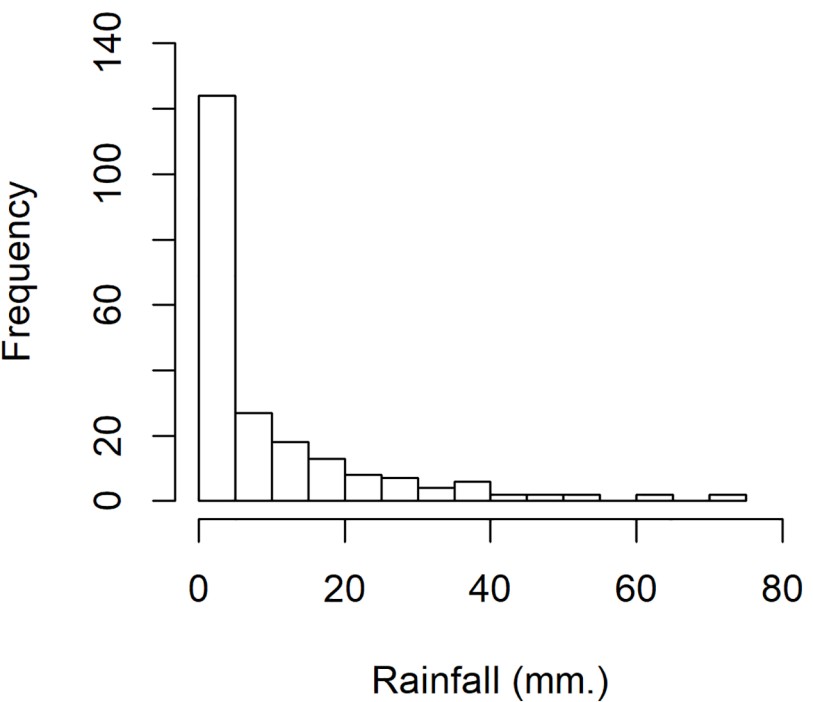

**Figure 1** The density of rainfall data in July 2015 for national parks in Nan province, Thailand.

**Table 3** AIC results to check the distributions of positive rainfall values in July 2015 for national parks in Nan province, Thailand.

| Densities | Normal | Lognormal | Cauchy | Exponential |
|---|---|---|---|---|
| AIC | 978.4592 | 906.9903 | 971.7420 | 908.4876 |

data series, there were 217 rainfall measurements, of which 117 were positive, showing a right-skewed distribution. The density of this data is presented in Fig. 1. Next, the minimum Akaike information criterion (AIC) was first to test the distribution of the positive rainfall data. The results in Table 3 reveal that the AIC value of the lognormal distribution was smallest, thus the distribution of this positive data series was the lognormal distribution. To validate the AIC test, a normal Q–Q plot for log-transformation data series is shown in Fig. 2. The distribution of zero values in this rainfall series coincided with the method mentioned in the "Methods" section for a binomial distribution. Therefore, a delta-lognormal distribution was appropriate for these data. Next, summary statistics were computed: $n = 217$, $\hat{\delta} = 0.5392$, $\hat{\mu} = 2.4762$, $\hat{\sigma}^2 = 0.9381$, and CV = 1.9337. Finally, the 95% confidence intervals for $\eta$ were calculated, as reported in Table 4. These results correspond with those from the simulation study when the sample size was large in that the coverage probabilities of the Bayesian methods (equitailed confidence intervals) were greater than the target. This indicates that the Bayesian method using the Jeffreys' Rule prior is appropriate to construct a confidence interval for this rainfall data due to it having the shortest expected length compared to the other methods. The estimated coefficient of variation in Table 4 means that the variability of the rainfall was rather high. This indicates
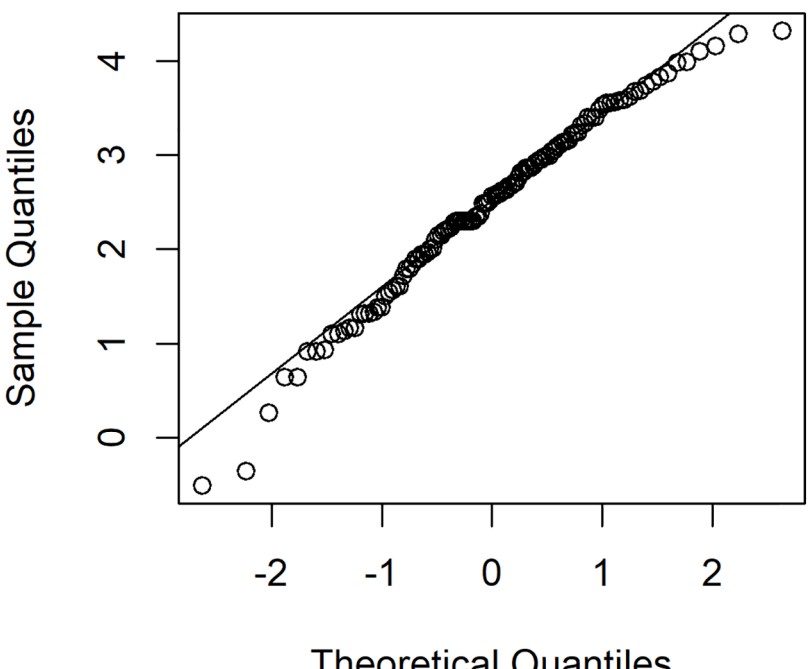

**Figure 2** The normal Q–Q plot of log-transformed for positive rainfall data in July 2015 for national parks in Nan province, Thailand.               

**Table 4** The 95% confidence intervals for a single coefficient of variation of rainfall data in July 2015 for national parks in Nan province, Thailand.

| Methods | Confidence intervals for η | | Length of intervals |
|---|---|---|---|
| | Lower | Upper | |
| Bayesian: The independent Jeffreys (Equitailed) | 1.6570 | 2.3579 | 0.7009 |
| Bayesian: The Jeffreys' Rule (Equitailed) | 1.6610 | 2.3460 | 0.6850 |
| Bayesian: The uniform (Equitailed) | 1.6646 | 2.3560 | 0.6914 |
| Bayesian: The independent Jeffreys (HPD) | 1.6314 | 2.3166 | 0.6852 |
| Bayesian: The Jeffreys' Rule (HPD) | 1.6424 | 2.3170 | 0.6746 |
| Bayesian: The uniform (HPD) | 1.6549 | 2.3345 | 0.6796 |
| FGCI | 1.6788 | 2.3294 | 0.6506 |

that the rainfall fluctuated, which would have affected the water levels in the area and there could have been flooding, which would have affected agricultural productivity in the area.

### Example 2

To investigate variation in rainfall, a rainfall dataset reported by the Upper Northern Region Irrigation Hydrology Center, Bureau of Water Management and Hydrology Royal Irrigation Department Thailand for August 2018 comprising eight precipitation stations in Nan province, Thailand (Muang, Thawangpha, Thung Chang, Pua, Song Khwae, Santisuk, Chaloem Phra Kiat, and Chiang Klang) was used. There were 248 observed values comprising 91 zero values and 157 positive values; the density of this rainfall data is shown in Fig. 3. The positive values follow a lognormal distribution, as indicated by the
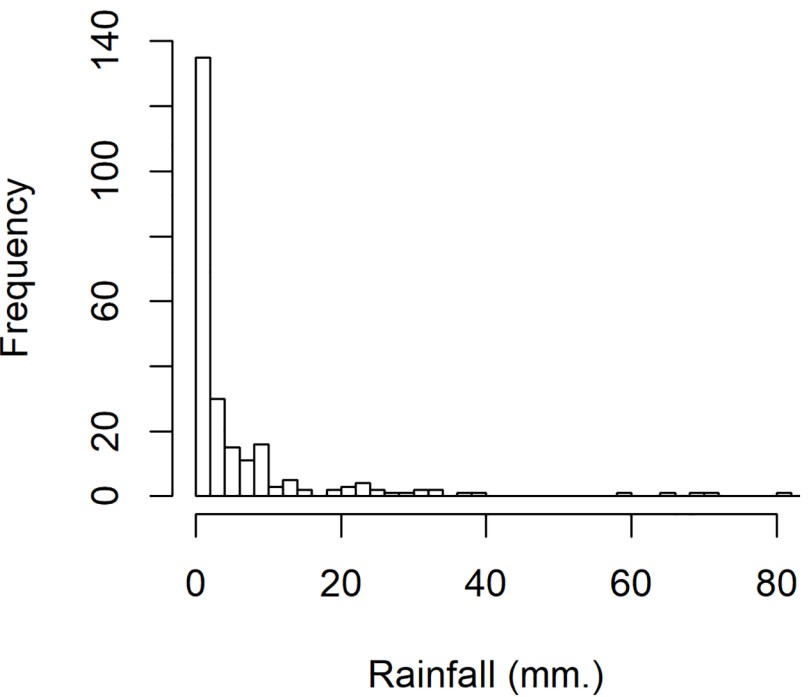

**Figure 3 The density of rainfall data in August 2018 from eight precipitation stations in Nan province, Thailand.**               

**Table 5 AIC results to check the distributions of positive rainfall values in August 2018 from eight precipitation stations in Nan province, Thailand.**

| Densities | Normal | Lognormal | Cauchy | Exponential |
|---|---|---|---|---|
| AIC | 1,596.0140 | 1,073.3380 | 1,196.0190 | 1,186.0920 |

minimum AIC in Table 5 and a normal Q–Q plot of the log-transformed data displayed in Fig. 4. In addition, the zero values have a binomial distribution (as discussed by *Aitchison (1955)*), thus the overall distribution is delta-lognormal. The summary statistics were $n = 248$, $\hat{\delta} = 0.6331$, $\hat{\mu} = 1.5822$, $\hat{\sigma}^2 = 2.2598$, and CV = 3.7595.

The results in Table 6 report the 95% confidence intervals for $\eta$. The results of the methods to construct the confidence intervals are in accordance with those in the simulation study for the case of a large sample size. The Bayesian method based on the Jeffreys' Rule prior (equitailed confidence intervals) had the shortest expected length. The coefficient of variation estimation in Table 6 indicates that the rainfall of this area was highly volatile, which affected the water level of the Nan River. Moreover, there might have been flooding in some of the areas due to high rainfall.

## DISCUSSION

Our findings reveal that the Bayesian method using the independent Jeffreys' prior to construct the equitailed confidence intervals performed well for all cases due to the coverage probabilities being consistently greater than or close to the nominal confidence level while the expected lengths were mostly no different from the other methods.

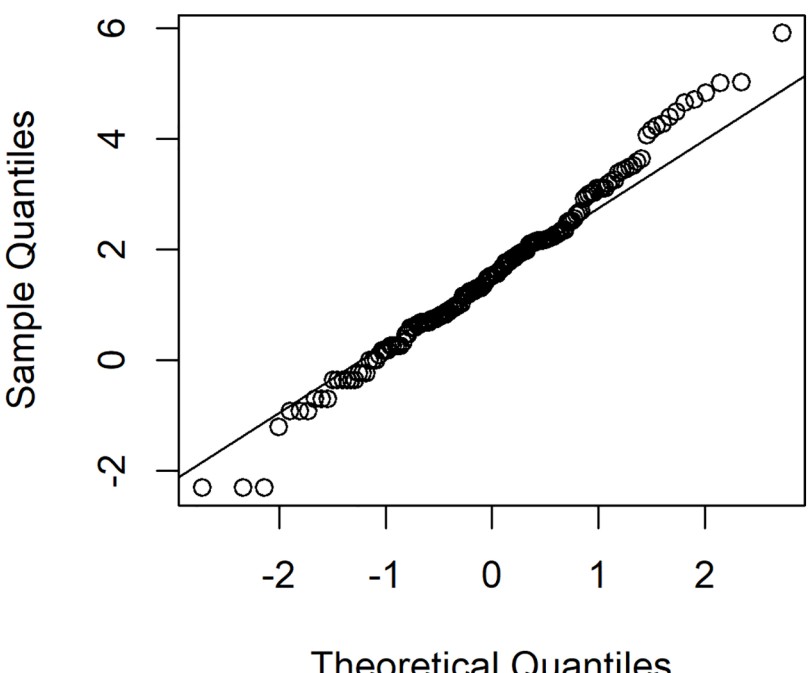

**Figure 4 The normal Q–Q plot of log-transformed for positive rainfall data in August 2018 from eight precipitation stations in Nan province, Thailand.**

**Table 6 The 95% confidence intervals for a single coefficient of variation of rainfall data in August 2018 from eight precipitation stations in Nan province, Thailand.**

| Methods | Confidence intervals for η | | Length of intervals |
|---|---|---|---|
| | Lower | Upper | |
| Bayesian: The independent Jeffreys (Equitailed) | 2.9429 | 5.1996 | 2.2567 |
| Bayesian: The Jeffreys' Rule (Equitailed) | 2.9536 | 5.1211 | 2.1675 |
| Bayesian: The uniform (Equitailed) | 2.9784 | 5.2196 | 2.2412 |
| Bayesian: The independent Jeffreys (HPD) | 2.7824 | 4.9280 | 2.1456 |
| Bayesian: The Jeffreys' Rule (HPD) | 2.8144 | 4.9014 | 2.0870 |
| Bayesian: The uniform (HPD) | 2.8704 | 5.0156 | 2.1452 |
| FGCI | 2.9795 | 5.1291 | 2.1496 |

Moreover, underestimation occurred for a few of the cases when applying the Bayesian methods based on the Jeffreys' Rule prior (equitailed), the independence Jeffreys' prior (HPD), and the uniform prior (HPD), and it appeared in almost all cases of the Jeffreys' Rule prior (HPD). In contrast, overestimation occurred in a few cases of applying the Bayesian method based on the uniform prior (equitailed) when the sample size was less than 100 together with a small variance and high proportion of non-zero values.

## CONCLUSIONS

We proposed the construction of confidence intervals for a single coefficient of variation of a delta-lognormal distribution using Bayesian methods and compared them with FGCI.

The Bayesian methods, which are based on the independent Jeffreys' prior, the Jeffreys' Rule prior, and the uniform prior, were constructed under equitailed confidence intervals or HPD intervals. The performance of the confidence intervals was assessed using the coverage probability and expected length through Monte Carlo simulations. The simulation studies showed that the Bayesian equitailed confidence intervals based on the independent Jeffreys' prior is recommended as a confidence interval for a single coefficient of variation. Future researchers may also be extended to the case of the coefficients of variation function.

### Funding
This research was funded by King Mongkut's University of Technology North Bangkok (Grant number: KMUTNB-61-PHD-004). The funders had no role in study design, data collection and analysis, decision to publish, or preparation of the manuscript.

### Grant Disclosures
The following grant information was disclosed by the authors:
King Mongkut's University of Technology North Bangkok: KMUTNB-61-PHD-004.

### Competing Interests
The authors declare that they have no competing interests.

### Author Contributions
- Noppadon Yosboonruang performed the experiments, analyzed the data, authored or reviewed drafts of the paper.
- Sa-aat Niwitpong conceived and designed the experiments, approved the final draft.
- Suparat Niwitpong contributed reagents/materials/analysis tools, prepared figures and/or tables.

### Data Availability
The raw data are available in Tables 7 and 8. R code is available in the Supplemental File.

### Supplemental Information
Supplemental information for this article can be found online at http://dx.doi.org/10.7717/peerj.7344#supplemental-information.

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
