# Peer review of "Measuring the dispersion of rainfall using Bayesian confidence intervals for coefficient of variation of delta-lognormal distribution: a study from Thailand"

_PeerJ, doi:10.7717/peerj.7344_

## Round 0.1 · original submission · Major Revisions

We now have received two review reports on your manuscript. Based on these review reports, I advise you to further revise your paper.

Reviewer 1 ·

Basic reporting

See the attached file

Experimental design

See the attached file

Validity of the findings

See the attached file

Additional comments

See the attached file

Annotated reviews are not available for download in order to protect the identity of reviewers who chose to remain anonymous.

Reviewer 2 ·

Basic reporting

This is an interesting statistical area with a good applciation.

Experimental design

Not applicable.

Validity of the findings

I think the findings are correct as much as I am able to assess this from outside.

Additional comments

I like the paper but have a few suggestions:

It would be good to talk right in the ebginning about the application data. Show that there is a zero-inflation which makes a lot of sense as there are these summer months Dec to April with almost no rainfall. Then show, for example using the distribution function, that the the delta-log-normal matches very well the EDF.

I have some questions w.r.t. the methodology starting page 3. You derive all posteriors in closed form, so CIs and HPDs should be easly calculable with a standard function. What is the purpose of the algorithms then? They seem a bit mysterious.

I understand that all Bayesian CIs do good. But why not use clssical confidence intervals as comparisons? I mena those you would get using the SEs derived from the Fisher information matrix?

Fianlly, it would be nice to gibe useres some well-documented R-code that they can use to derive the intervals using their data.

---

## Round 0.2 · Minor Revisions

Based on the review comments, I suggest that you moderately revise your paper before it be accepted for publication.

Reviewer 1 ·

Basic reporting

None

Experimental design

None

Validity of the findings

None

Additional comments

Thank you for your revision. However, I have a few question as follows.

1. I(delta', mu, sigma^2) in eq.(9) as you use right now is the Fisher information of the parameters in the delta-LN distribution. It can lead to misunderstanding. So you should change the symbol I() in eq.(9) to represent that it is derived from the Binomial.

2. Which step is applied the matrix in eq.(8)?

3. Thank you for giving the R-code. Why you separate the codes for FGCI, HPD, and Equitailed confidence intervals, not compute them using the same simulated data? This is because you need to compare them.

Reviewer 2 ·

Basic reporting

Revision is fine. However, you responded to my comment on classical CIs as follows:

"For classical confidence intervals, we have previously been applied the Fletcher
method to construct the confidence intervals which can be seen from Yosboonruang et al.
(2018) as references. That method does not work well in term of the coverage probability and
the expected length. Therefore, we do not used it to compare this proposed method."

But I see nothing of this nature in the actual text. It would be good as Motivation to say what are the classical approaches and that they fail (with a reference) and why you did the work present here.

Experimental design

ok

Validity of the findings

ok

Additional comments

see above

---

## Round 0.3 · accepted · Accept

After comparing your revised manuscript (version 2) with the review comments of the previous round, I have decided to accept your paper for publication.